# Multi-Stage Transcriptome Analysis Identifies Key Molecular Pathways for Soybean Under Phosphorus-Limited Conditions

**DOI:** 10.3390/ijms26178385

**Published:** 2025-08-28

**Authors:** Xiulin Liu, Sobhi F. Lamlom, Xueyang Wang, Chunlei Zhang, Fengyi Zhang, Kezhen Zhao, Rongqiang Yuan, Bixian Zhang, Honglei Ren

**Affiliations:** 1Soybean Research Institute of Heilongjiang Academy of Agriculture Sciences, Harbin 150086, China; liuxiulin1002@126.com (X.L.); sobhifaid@alexu.edu.eg (S.F.L.); hljsnkywxy@163.com (X.W.); zhangchunlei1989@yeah.net (C.Z.); ddszhangfy2019@163.com (F.Z.); zhaokz928@163.com (K.Z.); yrq18846121189@163.com (R.Y.); 2Plant Production Department, Faculty of Agriculture Saba Basha, Alexandria University, Alexandria 21531, Egypt; 3Institute of Biotechnology of Heilongjiang Academy of Agricultural Sciences, Harbin 150023, China

**Keywords:** soybean, phosphorus deficiency, transcriptome sequencing, differentially expressed genes (DEGs), phosphorus use efficiency

## Abstract

Phosphorus deficiency significantly limits soybean production across 74% of China’s arable land. This study investigated the molecular mechanisms enabling soybean to access insoluble phosphorus through transcriptome sequencing of the Heinong 48 variety across four developmental stages (Trefoil, Flower, Podding, and Post-podding). RNA-Seq analysis identified 2755 differentially expressed genes (DEGs), with 2506 up-regulated and 249 down-regulated genes. Notably, early developmental stages showed the most substantial transcriptional reprogramming, with 3825 DEGs in the Trefoil stage and 10,660 DEGs in the Flower stage, compared to only 523 and 393 DEGs in the Podding and Post-podding stages, respectively. Functional enrichment analysis revealed 44 significantly enriched GO terms in the Trefoil stage and 137 in the Flower stage, with 13 GO terms shared between both stages. KEGG pathway analysis identified 8 significantly enriched pathways in the Trefoil stage and 21 in the Flower stage, including key pathways related to isoflavonoid biosynthesis, alpha-linolenic acid metabolism, and photosynthesis. Among 87 differentially expressed transcription factors from 31 families, *bHLH* (8.08%), *bZIP* (7.18%), and *WRKY* (5.94%) were most prevalent. These findings provide genetic targets for developing soybean varieties with improved phosphorus acquisition capacity, potentially reducing fertilizer requirements and supporting more sustainable agricultural practices.

## 1. Introduction

Phosphorus (P) is an indispensable macronutrient and fundamental for plant growth and development. It is a key component of essential biomolecules and is critical in various cellular processes [1,2,3]. However, approximately 40% of the world’s arable land is deficient in available P [4], making low-P stress one of the most prevalent challenges in agricultural ecosystems [5]. This issue is particularly severe in China, where 74% of arable land is deficient in effective P, and over 30% requires P fertilization to meet crop nutrient demands [6]. As a nonrenewable resource, phosphorus is rapidly depleted. Its deficiency can lead to significant yield reductions of 30% to 40% in crops [7,8]. Therefore, improving the efficiency of P absorption and utilization in crops is essential to reduce reliance on natural resources and mitigate the environmental impacts of excessive P fertilizer use [9]. To adapt to persistent P deficiency, plants have developed various strategies, including modifications to root system architecture, increased acid phosphatase (ACP) activity, secretion of small-molecular organic acids, activation of P-responsive genes, and the formation of symbiotic relationships with mycorrhizal fungi [10,11]. Under P-deficient conditions, plants often develop longer, more slender roots, increase lateral root production, form cluster roots, and enhance root hair length and density [11,12,13]. Additionally, plants secrete organic acids such as citric acid, malic acid, and oxalic acid, which solubilize phosphate in the soil, converting it into a form that plants can absorb [14]. Acid phosphatases produced by plants further facilitate the release and conversion of phosphate, enhancing P uptake efficiency [15].

Soybean (*Glycine max* L.) represents one of the world’s most economically important legume crops, providing a critical source of protein and oil for human consumption, livestock feed, and industrial applications [16,17,18]. As the world’s fourth most widely cultivated crop, soybean production spans diverse geographic regions with varying soil phosphorus availability [19]. Enhancing phosphorus efficiency in soybeans offers significant potential benefits for global agricultural sustainability. Previous studies have highlighted notable genetic variation in phosphorus acquisition and utilization efficiency among soybean genotypes, indicating the existence of adaptive mechanisms that could be utilized for crop improvement [20,21,22]. Soybean’s response to phosphorus limitation involves general adaptive mechanisms shared with other plant species and legume-specific strategies. Like other plants, phosphorus-deficient soybeans typically exhibit altered root architecture, increased phosphatase activity, and enhanced expression of phosphate transporters [10,23]. However, as a legume capable of forming symbiotic relationships with nitrogen-fixing rhizobia, soybean faces unique challenges in balancing nitrogen and phosphorus acquisition, as nodulation and nitrogen fixation processes impose additional phosphorus demands [24,25].

Previous studies examining soybean responses to phosphorus limitation have primarily focused on specific tissues, developmental stages, or individual adaptive mechanisms. Zhang et al. [10] identified the acid-phosphatase-encoding gene *GmACP1* as a significant contributor to soybean phosphorus efficiency, demonstrating its role in enhancing phosphorus acquisition from organic sources. Similarly, Cai et al. [11] mapped the acid phosphatase gene *GmHAD1* to a quantitative trait locus associated with phosphorus efficiency. Peng et al. [2] characterized members of the *GmALMT* gene family, revealing their function in organic acid secretion under phosphorus limitation. Transcriptomic approaches have also been applied to understand soybean responses to phosphorus deficiency, but typically with limited temporal resolution or developmental context [26]. Recent advances in next-generation sequencing technologies provide unprecedented opportunities to comprehensively characterize transcriptional reprogramming in response to environmental stresses. RNA sequencing (RNA-Seq) enables the unbiased, genome-wide profiling of gene expression with high sensitivity and accuracy [27,28]. When applied across developmental stages, RNA-Seq can reveal the temporal dynamics of stress responses and identify stage-specific adaptive mechanisms that might be missed in single-time-point analyses. Furthermore, integrating transcriptomic data with functional annotation and pathway analysis can illuminate the biological processes and regulatory networks underlying complex adaptive traits such as phosphorus efficiency. The cellular and molecular mechanisms controlling phosphorus homeostasis involve complex regulatory networks encompassing phosphorus sensing, signal transduction, transcriptional reprogramming, and post-translational modifications [29,30]. Transcription factors play central roles in coordinating these responses, with members of the *MYB*, *WRKY*, *bHLH*, and *bZIP* families implicated in regulating phosphate starvation responses across various plant species [10,26].

This study aims to clarify the molecular mechanisms underlying soybean responses to insoluble phosphorus conditions across four critical developmental stages: Trefoil, Flowering, Podding, and Post-podding. We utilize comprehensive transcriptome sequencing of the Heinong 48 soybean variety to identify differentially expressed genes at these developmental stages and characterize their functional significance through Gene Ontology and pathway enrichment analyses.

## 2. Results

### 2.1. Different Patterns of Phosphorus Distribution in Soybean Organs

Analysis of phosphorus distribution across soybean plant organs revealed distinct allocation patterns under phosphorus stress conditions (Figure 1A,B). Pods demonstrated the highest phosphorus content among all plant organs, with 22.6% in control treatment and 23.2% in the phosphorus-limited treatment. This preferential allocation to reproductive organs was consistently observed across both experimental conditions, indicating a robust biological response independent of phosphorus availability levels.

Vegetative organs showed markedly lower phosphorus content compared to pods. Roots contained 10.6% (control) and 11.9% (treatment) phosphorus, while leaves maintained 7.2% (control) and 9.5% (treatment). The stem’s phosphorus content was the lowest among all organs, at 5.2% in control and 4.3% in treatment conditions (Figure 1B). The heatmap visualization confirmed this hierarchical distribution pattern, with pods displaying the most intense coloration corresponding to the highest phosphorus concentration (Figure 1B). The differential allocation pattern reflects the plant’s reproductive priority strategy, where phosphorus is actively mobilized from vegetative tissues to support seed development. This redistribution ensures adequate phosphorus availability for energy-intensive processes during pod filling, including protein synthesis, oil accumulation, and phytic acid storage.

### 2.2. Transcriptome Sequencing and Quality Assessment

RNA-Seq analysis was performed on soybean leaf samples collected during four developmental stages under low-phosphorus stress: Trefoil (A and B), Flower (C and D), Podding (E and F), and Post-podding (G and H). In this study, A, C, E, and G are control samples, whereas B, D, F, and H are treatment samples. A total of 24 leaf RNA libraries were sequenced, generating an average of 1.4 billion 150 bp paired-end reads (ranging from 45.3 to 76.71 million reads per sample) (Appendix A). After quality filtering, approximately 1.4 billion clean reads were obtained, with GC content ranging from 44.66% to 48.77% (Appendix A). On average, 93.88% of high-quality reads (Q ≥ 30) mapped to the soybean reference genome, with mapping rates from 56.50% to 94.19% (Appendix A). Principal component analysis (PCA) revealed clear differences in gene expression patterns that are clustered by developmental stage (Figure 2), with similar expression profiles observed between control and treatment samples within each stage.

### 2.3. Differentially Expressed Genes Across Developmental Stages

The analysis of differential expression revealed differing numbers of DEGs between treatment and control conditions at the four developmental stages (Figure 3a). We identified 3825 DEGs in the Trefoil stage (B_vs_A), with 2265 up-regulated and 1560 down-regulated, 10,660 DEGs in the Flower stage (D_vs_C), with 5644 up-regulated and 5016 down-regulated, 523 DEGs in the Podding stage (F_vs_E), with 159 up-regulated and 364 down-regulated, and 393 DEGs in the Post-podding stage (H_vs_G), consisting of 259 up-regulated and 134 down-regulated. Out of a total of 15,401 DEGs identified across all stages, 25% were found in the Trefoil stage, 69% in the Flower stage, and 3% each in the Podding and Post-podding stages, which suggests that low-phosphorus treatment mainly impacts early developmental stages. Further scrutiny of significant DEGs, using rigorous filtering criteria (|log2FC| ≥ 11 and FDR < 0.01), revealed 39 highly differential genes (Figure 2 Group B). The log2FC values spanned from −11.57 to 14.11 in the Trefoil stage, −14.67 to 12.17 in the Flower stage, −10.90 to 11.06 in the Podding stage, and from −11.11 to 10.31 in the Post-podding stage. K-means clustering of DEGs at each developmental stage displayed two distinct expression patterns for both treatment and control groups (Figure 2 Group C–F), verifying the differential response to low-phosphorus stress throughout development stages.

### 2.4. Functional Enrichment Analysis of DEGs

Since 94% of DEGs were concentrated in the Trefoil and Flower stages, we focused our functional characterization on these early developmental periods. Gene Ontology (GO) enrichment analysis revealed 44 significantly enriched terms in the Trefoil stage, with 21 terms enriched in down-regulated DEGs and 23 terms in up-regulated DEGs (Figure 4a). Using stringent criteria (*p*.adjust < 0.001 and log2FoldEnrichment > 1.5), we identified four key GO terms: “anchored component of membrane” (GO:0031225), “plant-type cell wall” (GO:0009505), “anchored component of plasma membrane” (GO:0046658), and “plant-type cell wall organization or biogenesis” (GO:0071669). In the Flower stage, 137 GO terms were significantly enriched, with 55 terms associated with down-regulated DEGs and 82 with up-regulated DEGs. Nineteen key GO terms met our stringent criteria, including five terms enriched in down-regulated DEGs related to glucosyltransferase activity, response to ozone, and oxidoreductase activity. The remaining 14 terms, enriched in up-regulated DEGs, were predominantly associated with chloroplast thylakoid, plastid thylakoid, thylakoid membrane, photosynthetic membrane, and cuticle development. KEGG pathway analysis identified eight significantly enriched pathways in the Trefoil stage (two associated with down-regulated DEGs, six with up-regulated DEGs) and twenty-one pathways in the Flower stage (fifteen associated with down-regulated DEGs, six with up-regulated DEGs) (Figure 4b). Key pathways (*p*.adjust < 0.001 and log2FoldEnrichment > 1.5) included “Protein processing in endoplasmic reticulum” (ko04141) and “Phagosome” (ko04145) in the Trefoil stage and “Isoflavonoid biosynthesis” (ko00943), “alpha-Linolenic acid metabolism” (ko00592), “Butanoate metabolism” (ko00650), “Photosynthesis” (ko00195), and “Cutin, suberine and wax biosynthesis” (ko00073) in the Flower stage. We identified thirteen GO terms and three KEGG pathways that were significantly enriched (*p*.adjust < 0.05) in both the Trefoil and Flower stages (Figure 4c). The common GO terms were primarily related to cell wall organization, biogenesis, and metabolism, while the shared KEGG pathways included “Phagosome” (ko04145), “Motor proteins” (ko04814), and “Amino sugar and nucleotide sugar metabolism” (ko00520).

### 2.5. Transcription Factor Dynamics Under Low-Phosphorus Stress

We annotated 73,791 genes using the Plant Transcription Factor Database (PlantTFDB 5.0), identifying 87 transcription factors (TFs) that were differentially expressed under low-phosphorus stress. The representation of these TFs ranged from 0.03% to 8.08% of the total transcription factors. The 31 TF families with representation > 1% accounted for 80.65% of all differentially expressed TFs (Figure 5a), with the top 10 families being *bHLH* (8.08%), *bZIP* (7.18%), *WRKY* (5.94%), *MYB* (5.41%), *MYB*_related (4.67%), *ERF* (4.61%), *C2H2* (3.90%), *C3H* (3.09%), *ARF* (2.51%), and NAC (2.48%) (Figure 5b). In the Trefoil stage, eight TFs showed significant differential expression, with three up-regulated (*Dof*, *GeBP*, and *OFP*) and five down-regulated (*CAMTA*, *HB-other*, *RB*, *SNF2*, and *TAZ*) (Figure 5c). The Flower stage exhibited more dramatic changes, with 31 TFs showing significant differential expression, with 18 up-regulated (including *ARID*, *BES1*, *bHLH*, *bZIP*, *C2H2*, and others) and 13 down-regulated (including *B3*, *CAMTA*, *ERF*, *NAC*, and others) (Figure 5d).

## 3. Discussion

Our transcriptome analysis reveals a striking stage-dependent response of soybean to low-phosphorus stress, with the most pronounced transcriptional reprogramming occurring during early developmental stages. The significantly higher number of DEGs identified in the Trefoil (3825) and Flower (10,660) stages compared to the Podding (523) and Post-podding (393) stages suggests that soybean plants establish their adaptive mechanisms to phosphorus deficiency primarily during early development. This finding aligns with previous studies in various crops that have demonstrated heightened sensitivity to nutrient stress during vegetative and early reproductive phases [31,32]. The substantial increase in DEGs during the transition from the Trefoil to Flower stage (from 25% to 69% of total DEGs) indicates a critical developmental window where phosphorus availability significantly impacts gene expression. This pattern may represent an evolutionary strategy where plants allocate resources to establish robust phosphorus acquisition and utilization mechanisms before transitioning to reproductive stages, thereby ensuring sufficient phosphorus reserves for seed development [33,34,35]. The reduced transcriptional response during later developmental stages could reflect either successful adaptation to the stress condition or a shift in resource allocation priorities toward reproductive processes [36,37].

The significant enrichment of GO terms related to cell wall components and organization in the Trefoil stage, particularly “anchored component of membrane” (GO:0031225), “plant-type cell wall” (GO:0009505), and “plant-type cell wall organization or biogenesis” (GO:0071669), suggest that cell wall remodeling is an early response to phosphorus limitation. Cell wall modifications have been previously implicated in nutrient stress responses, as they can alter root architecture, increase root hair density, and enhance root exudation to improve phosphorus acquisition [38,39,40]. In the Flower stage, the enrichment of GO terms associated with chloroplast thylakoid, plastid thylakoid, thylakoid membrane, and photosynthetic membrane in up-regulated DEGs indicates a potential enhancement of photosynthetic capacity. This transcriptional adjustment may represent a compensatory mechanism to maintain energy production under phosphorus-limited conditions, which typically impair ATP synthesis and metabolism. The increased expression of photosynthesis-related genes could also reflect a reallocation of resources from phosphorus-rich metabolites to carbon assimilation [41,42].

The KEGG pathway analysis revealed significant enrichment of “Isoflavonoid biosynthesis” (ko00943) and “alpha-Linolenic acid metabolism” (ko00592) pathways in the Flower stage, suggesting a shift in secondary metabolite production under phosphorus stress. Isoflavonoids play crucial roles in plant defense and root nodulation, while alpha-linolenic acid serves as a precursor for jasmonic acid, a key signaling molecule in stress responses [43,44]. The induction of these pathways may enhance root–microbe interactions, particularly with phosphate-solubilizing microorganisms, thereby improving phosphorus acquisition from the soil. The significant enrichment of “Protein processing in endoplasmic reticulum” (ko04141) in the Trefoil stage suggests alterations in protein folding, quality control, and degradation processes. This could reflect the plant’s effort to optimize protein synthesis and turnover under phosphorus limitation, potentially directing resources toward essential proteins involved in phosphorus acquisition and utilization [45,46].

Identifying 87 differentially expressed transcription factors (TFs) spanning 31 families provides insights into the regulatory networks controlling the phosphorus deficiency response. The predominance of the *bHLH* (8.08%), *bZIP* (7.18%), and WRKY (5.94%) families among the differentially expressed TFs is particularly noteworthy, as these families have been implicated in nutrient stress responses in various plant species. The regulation of root morphology is a topic that has been extensively researched and is influenced by multiple signaling pathways. The AUX signal-related genes OsARF12 and OsTOP1 facilitate primary root growth in rice [47,48]. Genes associated with SA signaling, such as OsAIM1, and GA signaling, including *OsSHB*, contribute to the development of primary roots [49,50]. The MYB-type transcription factor OsMYB1 is implicated in lateral root growth. Root hairs are crucial for phosphorus (P) absorption in plants [51]. In Arabidopsis, various transcription factors (TFs) are implicated in root hair development, including MYB types such as *MEMBRANE ANCHORED MYB* (*maMYB*) [52], *bHLH types like RHD SIX-LIKE 1* (*RSL1*) [53] and *RHD SIX-LIKE 3* (*RSL3*) [54], Lotus japonicus *ROOT HAIR LESS-LIKE* 2 (LRL2) [55], and *HD-ZIP* types such as *HOMEODOMAIN GLABROUS 11* (*HDG11*) [56]. Additionally, genes activated by phosphorus deficiency that facilitate root hair formation have been identified. *OsAUX1*, which encodes an auxin influx transporter, facilitates root hair extension under low-phosphorus stress in rice [57]. In Arabidopsis, the ethylene signaling gene EIN3 and its closely related homolog EIL1 have been shown to play a role in root hair formation triggered by phosphorus deficiency [58].

In the Trefoil stage, the up-regulation of *Dof*, *GeBP*, and *OFP* transcription factors suggests their involvement in early signaling of phosphorus deficiency. *Dof* TFs, in particular, have been shown to regulate the expression of phosphate transporters and phosphorus homeostasis genes in Arabidopsis [59]. The down-regulation of *CAMTA*, *HB*-other, *RB*, *SNF2*, and *TAZ* TFs may indicate a repression of growth-promoting pathways to conserve phosphorus resources. The more complex TF dynamics observed in the Flower stage, with 18 up-regulated and 13 down-regulated TF families, likely reflect the broader transcriptional reprogramming occurring during this critical developmental transition. The up-regulation of *bHLH*, *bZIP*, and *C2H2* TFs may drive the induction of phosphate acquisition mechanisms, while the down-regulation of *ERF* and *NAC* TFs could modulate developmental processes to prioritize phosphorus conservation.

In conclusion, our transcriptome analysis provides valuable insights into the molecular mechanisms underlying soybean adaptation to phosphorus limitation across various developmental stages, thereby advancing our understanding of nutrient stress responses in crops and offering potential pathways for sustainable agricultural systems in phosphorus-limited environments.

## 4. Materials and Methods

### 4.1. Plant Material and Growth Conditions

Soybean variety Heinong 48, provided by the Soybean Research Institute of Heilongjiang Academy of Agricultural Sciences, was used in this study. To avoid interference from medium-derived phosphorus, plants were cultivated in washed vermiculite. For the phosphorus-deficient condition, 10 g of insoluble phosphorus Ca_10_(PO_4_)(OH)_5_ was thoroughly mixed with 15 kg of vermiculite, while no phosphorus was added to the control condition. Plants were grown in a controlled environment chamber at 28 °C/22 °C (day/night) with a 12 h/12 h photoperiod and 80% relative humidity.

### 4.2. Sample Collection and Preparation

Healthy young roots were harvested at the three-leaf stage in August 2024. For developmental stage analysis, leaf samples (0.1 g) were collected at four critical developmental stages, Trefoil (A and B), Flowering (C and D), Podding (E and F), and Post-podding (G and H), under phosphorus-deficient conditions. All samples were immediately flash-frozen in liquid nitrogen. Three biological replicates were collected for each treatment, resulting in a total of twenty-four samples. The trefoil (three-leaf) stage was selected as representing a critical early vegetative phase when phosphorus demand is high for root establishment and leaf expansion, making this developmental window particularly responsive to phosphorus availability and optimal for capturing early transcriptional responses. Our multi-stage sampling approach across four developmental phases (Trefoil, Flowering, Podding, Post-podding) was designed to comprehensively characterize phosphorus stress responses throughout soybean development, capturing key metabolic transitions from vegetative growth to reproductive maturity when phosphorus allocation priorities shift toward seed development.

### 4.3. RNA Extraction and Quality Control

Total RNA was extracted using TRIzol reagent (Invitrogen, CA, USA) according to the manufacturer’s protocol. RNA integrity was assessed by 1.2% agarose gel electrophoresis, and concentration was measured using an Agilent 2100 Bioanalyzer (Agilent Technologies, Santa Clara, CA, USA). Only RNA samples with RNA Integrity Number (RIN) values greater than 7.0 were used for subsequent analyses.

### 4.4. Library Construction and RNA Sequencing

Following the manufacturer’s recommendations, RNA-seq libraries were constructed using the NEBNext Ultra™ RNA Library Prep Kit for Illumina (NEB, Beijing, China). The prepared libraries were sequenced on an Illumina NovaSeq 6000 platform (Illumina Inc., California, USA) to generate paired-end reads. The clean reads were mapped to the soybean reference genome (GCF_004193775.1, https://www.ncbi.nlm.nih.gov/datasets/genome/GCF_004193775.1 accessed on 25 May 2025) using the STAR v2.7.10a software. The sequence data were deposited in the NCBI Sequence Read Archive under BioProject ID: PRJNA1232401. Transcript abundance was quantified as Fragments Per Kilobase of transcript per Million mapped reads (FPKM) using RNA-Seq by Expectation-Maximization (RSEM v1.3.2). Differentially expressed genes (DEGs) were identified using the edgeR package in R with a threshold of false discovery rate (FDR) < 0.05 and *p*-value < 0.05.

### 4.5. Functional Enrichment Analysis

Gene Ontology (GO) and Kyoto Encyclopedia of Genes and Genomes (KEGG) pathway enrichment analyses of DEGs were performed using clusterProfiler v4.8.2 with the Benjamini–Hochberg correction. Pathways with adjusted *p*-values < 0.05 were considered significantly enriched. The fold enrichment was calculated as the ratio of gene frequency in the test set (GenRatio) to gene frequency in the reference set (BgRatio). Differential transcription factor analysis was conducted using the STAMP software, with FDR < 0.05 and *p*-value < 0.05 as the threshold for identifying differentially expressed transcription factors.

### 4.6. Statistical Analysis

A one-way analysis of variance, followed by Tukey’s test, was conducted to determine statistically significant differences among groups and tissues (*p* < 0.05). Additionally, a Spearman two-tailed test was used to determine significant correlations (*p* < 0.05) between growth rates within the plant samples. Nonetheless, no statistically significant difference was observed in the growth rate between the high-phosphorus-stress and normal-phosphorus groups.

## 5. Conclusions

Transcriptome analysis reveals the remarkable molecular adaptability of soybean to phosphorus limitation across developmental stages. The Heinong 48 variety exhibits 2755 differentially expressed genes under phosphorus deficiency, with 94% concentrated in the Trefoil and Flower stages, having a dramatic impact on gene expression. GO and KEGG pathway analyses identified key biological processes responding to phosphorus limitation, including early cell wall remodeling in the Trefoil stage and enhanced photosynthetic capacity and secondary metabolite pathways during flowering. These adaptations suggest strategic metabolic reprogramming to maintain development under phosphorus shortage. The identification of 87 differentially expressed transcription factors illuminates predominantly from the *bHLH*, *bZIP*, and *WRKY* families, shedding light on the regulatory networks controlling phosphorus deficiency responses. The differential expression patterns of these transcription factors indicate sequential activation of adaptive mechanisms, with distinct regulatory programs during early vegetative and reproductive development. The 13 GO terms and 3 KEGG pathways enriched in both the Trefoil and Flower stages represent fundamental adaptation processes established during early development. These findings suggest that breeding strategies to improve phosphorus utilization efficiency should focus on traits expressed during early developmental stages. By elucidating stage-specific transcriptional reprogramming, metabolic adjustments, and regulatory networks, this study provides valuable genetic targets for developing phosphorus-efficient soybean cultivars, which may potentially reduce phosphorus fertilizer requirements and support more sustainable agricultural practices in phosphorus-limited environments. Building on our transcriptomic foundation, future research will prioritize multi-omics integration by combining metabolomics and proteomics to validate gene expression changes at the protein and metabolite levels, while conducting functional validation studies through gene knockout/overexpression experiments for key identified transcription factors (*bHLH*, *MYB*, *WRKY* families). We plan to translate these molecular insights into practical breeding applications by developing molecular markers for phosphorus use efficiency genes and implementing marker-assisted selection strategies. Additionally, field validation studies will evaluate our findings under diverse soil phosphorus conditions and environmental stresses, while mechanistic investigations will explore epigenetic regulation, root–microbiome interactions, and phosphorus signaling network crosstalk with other nutrient stress pathways to develop phosphorus-efficient cultivars suitable for sustainable low-input agricultural systems.

## Figures and Tables

**Figure 1 ijms-26-08385-f001:**
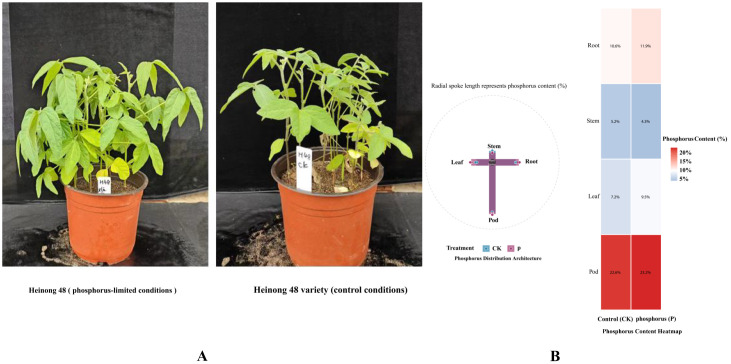
Tissue-specific phosphorus allocation across plant organs in response to nutrient treatments. (**A**) Two potted plants labeled Heinong 48-P and Heinong 48-CK. (**B**) Control (CK)= phosphorus-deficient treatment (pure vermiculite); P = phosphorus-limited treatment (with Ca_10_(PO_4_)(OH)_5_). The radial spoke length in the polar plot represents the percentage of phosphorus content in each organ. Phosphorus distribution analysis across soybean plant organs (cv. Heinong 48) showing preferential allocation patterns under two phosphorus stress conditions.

**Figure 2 ijms-26-08385-f002:**
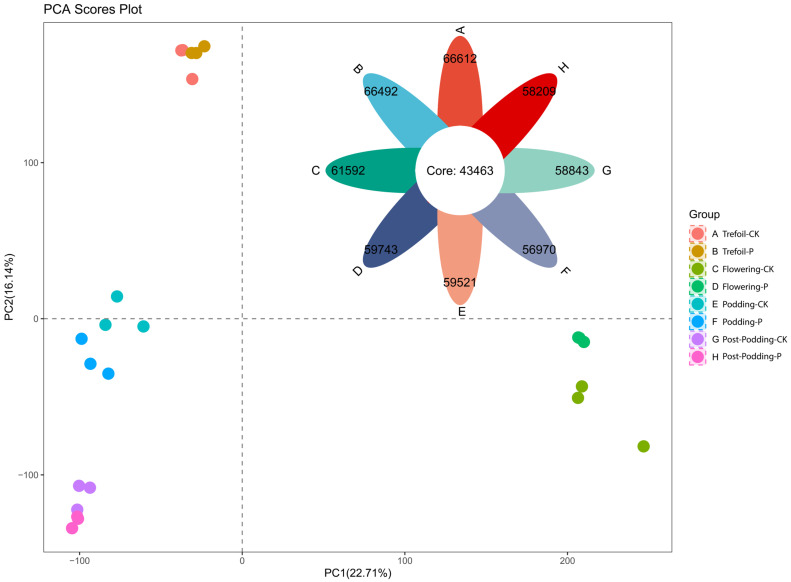
Transcriptomic variation across soybean developmental stages under phosphorus stress. Legend: Trefoil—control (A), Trefoil—treatment (B), Flowering—CK (C), Flowering—P (D), Podding—CK (E), Podding—P (F), Post-podding—CK (G), Post-podding—P (H). CK (control) = phosphorus-deficient (pure vermiculite); P = phosphorus-limited (with Ca_10_(PO_4_)(OH)_5_). Principal component analysis of gene expression profiles from soybean leaf samples across four developmental stages under two phosphorus stress conditions. Each point represents a biological replicate (n = 3 per group), with PC1 and PC2 explaining 22.71% and 16.14% of variance, respectively. Developmental stage drives primary transcriptomic separation along PC1, while phosphorus treatment effects are secondary. Tight clustering within groups demonstrates consistent gene expression patterns, with developmental progression creating greater transcriptomic differences than phosphorus treatment intensity.

**Figure 3 ijms-26-08385-f003:**
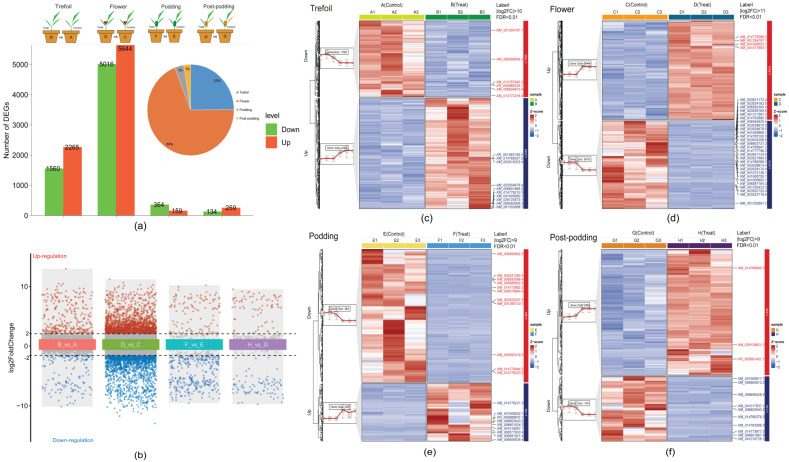
Identification of key genes in response to low-phosphorus stress in four periods. (**a**) Differentially expressed genes (DEGs) at different stages (Trefoil, Flower, Podding, and Post-podding) between the control and treatment under low-phosphorus stress. (**b**) Analysis of differential gene expression indicating up- and down-regulated genes across all four comparison groups. A log2FC > 1 and FDR < 0.05 is represented as significantly up-regulated in red, while a log2FC < −1 and FDR < 0.05 is represented as significantly down-regulated in green. A total of 39 key genes were identified at the extremes of the volcano plot with |log2FC| > 11 and FDR < 0.01. (**c**–**f**) The expression pattern of the DEGs across the four stages shown by a clustering heatmap: Trefoil (**c**), Flower (**d**), Podding (**e**), and Post-podding (**f**).

**Figure 4 ijms-26-08385-f004:**
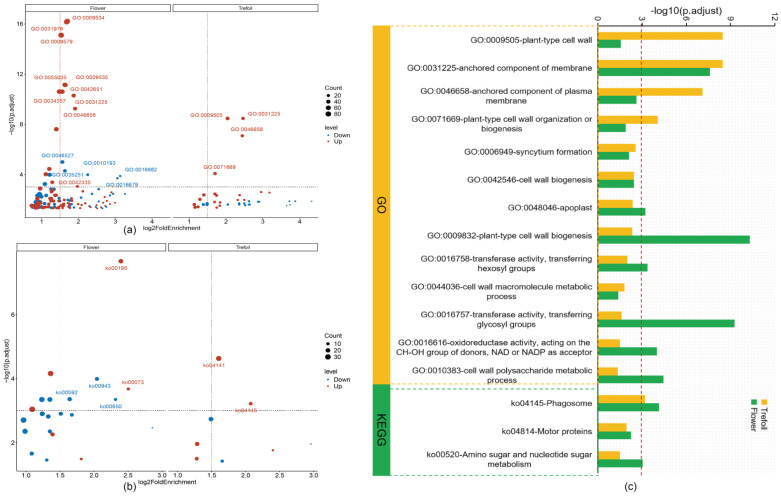
Functional characteristics of DEGs across different periods. (**a**) GO enrichment analysis of DEGs (up-regulated and down-regulated) during the Trefoil and Flower periods based on the GO database. The log2 (fold enrichment) is plotted on the x-axis, and −log10 (*p*.adjust) is plotted on the y-axis. The blue bubbles indicate down-regulated DEGs, while the red bubbles indicate up-regulated DEGs; the area of each bubble represents the number of genes enriched in the GO terms. Key terms are marked where *p*.adjust < 0.001 and log2FoldEnrichment > 1.5. (**b**) KEGG enrichment analysis of DEGs (up-regulated and down-regulated) during the Trefoil and Flower periods based on the KEGG database. The log2 (fold enrichment) is plotted on the x-axis, and −log10 (*p*.adjust) is plotted on the y-axis. The blue bubbles represent pathways enriched with down-regulated DEGs, while the red bubbles represent pathways enriched with up-regulated DEGs; again, the bubble area reflects the number of genes enriched in the KEGG pathways. Key pathways are marked where *p*.adjust < 0.001 and log2FoldEnrichment > 1.5. (**c**) The GO terms and KEGG pathways were significantly enriched in both the Trefoil and Flower periods. The x-axis shows the terms and pathways enriched in Trefoil (yellow) and Flower (green), represented by −log2 (*p*.adjust) with a threshold of *p* = 0.001. A total of 3 pathways and 13 terms were significantly enriched in both the Trefoil and Flower stages with *p*.adjust < 0.05.

**Figure 5 ijms-26-08385-f005:**
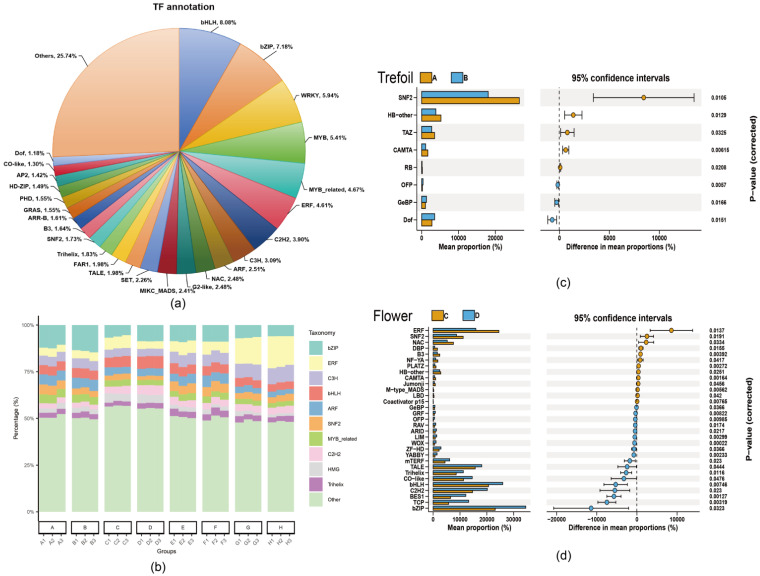
Transcription factor analysis across developmental stages under phosphorus stress. (**a**) Total of 87 transcription factors were annotated using PlantTFDB, with *bHLH* family most abundant (8.08%). (**b**) Distribution of top 10 TF families across four developmental stages (Trefoil, Flowering, Podding, Post-podding). (**c**) Trefoil stage: 8 differentially expressed TFs (3 up-regulated, 5 down-regulated) in phosphorus-limited treatment B vs. control A. (**d**) Flowering stage: 31 differentially expressed TFs (18 up-regulated, 13 down-regulated) in treatment D vs. control C, indicating major transcriptional reprogramming during reproductive transition. FDR < 0.05, |log_2_FC| > 1. The increased TF response from Trefoil (8 TFs) to Flowering (31 TFs) demonstrates escalating transcriptional complexity under phosphorus stress during critical developmental transitions.

## Data Availability

The original contributions presented in the study are included in the article/Appendix A; further inquiries can be directed to the corresponding author.

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
