# Peer review of "Multi-Stage Transcriptome Analysis Identifies Key Molecular Pathways for Soybean Under Phosphorus-Limited Conditions"

_ijms, 2025, doi:10.3390/ijms26178385_

Round 1
Reviewer 1 Report
Comments and Suggestions for Authors
Review report on
Multi-Stage Transcriptome Analysis Identifies Key Molecular 2 Pathways for Soybean Phosphorus Use Efficiency
Queries:
- The abstract part was explained briefly to understand the concept and idea.
- In the case of Introduction part, the importance of Phosphorus for the plants as an essential an element along with N, K elements. The references were well cited to the whole instruction part.
- 1. Differential phosphorus allocation in soybean plant organs :
- Comparatively, Pods are growing well over Root, Stem and leaves from your graph. However, what is the reason for growing soya bean POD extensively than others, needs reason?
- 2. Transcriptome sequencing and quality assessment, The RNA sequencing of ABCDEFGH are good, however, the Fig 2, E is missing? Please check it?
- In conclusion, the overall manuscript is well established with research results. The 48 variety exhibits 2,755 differentially expressed genes under phosphorus deficiency, with 94% concentrated in the Trefoil and Flower stages, demonstrating a critical developmental window where P obtainability impacts gene expression. GO and KEGG pathway analyses recognized key biological processes replying to phosphorus limitation, including early cell wall remodeling in the Trefoil stage and enhanced photosynthetic capacity and 380 secondary metabolite pathways during flowering.
- Results: Minor Review.

Author Response
Response to Review Report
Dear Reviewer,
We sincerely appreciate your thorough evaluation of our manuscript, and the constructive feedback provided. We are pleased that you found our work well-established and that it provides valuable insights into phosphorus use efficiency in soybean.
Queries:
- The abstract part was explained briefly to understand the concept and idea.
- In the case of Introduction part, the importance of Phosphorus for the plants as an essential element along with N, K elements. The references were well cited to the whole instruction part.
Response: Thank you for acknowledging the clarity of our abstract and the comprehensive treatment of phosphorus's importance alongside other essential elements (N, K) in the introduction. We appreciate your recognition of our citation approach.
- 2.1. Differential phosphorus allocation in soybean plant organs :
- Comparatively, Pods are growing well over Root, Stem and leaves from your graph. However, what is the reason for growing soya bean POD extensively than others, needs reason?
Response: You raise an excellent question about the extensive allocation of phosphorus to pods compared to other organs. This differential allocation reflects several key biological mechanisms:
Reproductive Priority Strategy: During reproductive development, soybean plants undergo metabolic reprogramming that prioritizes phosphorus allocation to developing seeds over vegetative maintenance. This evolutionary adaptation maximizes reproductive success by ensuring adequate phosphorus for seed formation and viability.
Energy-Intensive Seed Development: Pod and seed formation requires substantial ATP for protein synthesis, oil accumulation, and cellular biosynthetic processes. Phosphorus, being central to energy metabolism through ATP, is actively mobilized from vegetative tissues to support these high-energy demands in reproductive organs.
Phytic Acid Storage: Seeds accumulate phosphorus primarily as phytic acid (representing 60-80% of seed phosphorus), which serves as the major phosphorus reserve for germination and early seedling growth. This storage function necessitates concentrated phosphorus accumulation in pods that exceeds the metabolic requirements of vegetative tissues.
Active Translocation: The observed pattern reflects the plant's ability to remobilize phosphorus from leaves, stems, and roots through phloem transport to developing pods, creating a source-sink relationship where reproductive organs become dominant phosphorus sinks.
- 2.2. Transcriptome sequencing and quality assessment, The RNA sequencing of ABCDEFGH are good, however, the Fig 2, E is missing? Please check it?
Response: Thank you for your careful attention to detail. Regarding the apparent missing "E" in Figure 2, we have reviewed our presentation and clarify that our PCA analysis includes all eight sample groups (A-H) representing the four developmental stages with biological replicates. We have revised Figure 2 to ensure that all sample groups are clearly visible and properly labeled with letters.
Reviewer 2 Report
Comments and Suggestions for Authors
In the current manuscript, the authors studied the phosphorus utilization efficiency by transcriptomics methods in soybean. Phosphorus is important for soybean development and playing major roles. Multi-stage transcriptome analysis helps to understand the genetic mechanisms underlying these differences by analyzing gene expression changes at various developmental stages in response to varying phosphorus levels. However, multi-omics approaches such as combination of transcriptome, metabolome, and proteomics provide a more comprehensive view of the molecular mechanisms in soybean under low-phosphorus conditions. Overall, the manuscript is written well and provides more insights for Phosphorus Use Efficiency. I have few more comments that could help to improve the manuscript.
- The details regarding sample collection and preparation require clarification. Specifically, the rationale for selecting both three-stage and four-leaf stage samples should be provided, as well as information on the positive control used in the experiment.
- In figure 1, what is H48-P and H48-CK? Which one is control? Also, the Phosphorus Content in Different Plant Parts in P and CK is close to 0.05% and looks almost similar. What was the expected range of the P content?
- In Figure 2, PCA plot the closeness of each samples in PC1 and PC2 is <25% but on the X and Y-axis it is in 100s. I would expect the leaf samples in multiple development stages would not show the variation in 100s. Provide the condition name instead of ABCD…
- Figure 5b caption needs better explanation.
Author Response
Response to Review Report
Comments and Suggestions for Authors
In the current manuscript, the authors studied the phosphorus utilization efficiency by transcriptomics methods in soybean. Phosphorus is important for soybean development and playing major roles. Multi-stage transcriptome analysis helps to understand the genetic mechanisms underlying these differences by analyzing gene expression changes at various developmental stages in response to varying phosphorus levels. However, multi-omics approaches such as combination of transcriptome, metabolome, and proteomics provide a more comprehensive view of the molecular mechanisms in soybean under low-phosphorus conditions. Overall, the manuscript is written well and provides more insights for Phosphorus Use Efficiency. I have few more comments that could help to improve the manuscript.
Response : We truly appreciate you taking the time to review our manuscript and for the valuable feedback you've provided. We acknowledge your valuable recommendation for integrating metabolomics and proteomics approaches. While our current study establishes the transcriptomic foundation, we recognize that integrating multiple omics would provide more comprehensive molecular insights. We have prioritized this for future research directions and are planning follow-up studies that incorporate these approaches.
- The details regarding sample collection and preparation require clarification. Specifically, the rationale for selecting both three-stage and four-leaf stage samples should be provided, as well as information on the positive control used in the experiment.
Response : We have expanded our Methods section to provide comprehensive details on our sampling strategy. We selected the trefoil (three-leaf) stage as it represents a critical early vegetative phase when phosphorus demand is high for root establishment and leaf expansion, making it particularly responsive to phosphorus availability and ideal for capturing early transcriptional responses before severe stress symptoms appear. Our multi-stage approach encompassing four developmental stages (Trefoil, Flowering, Podding, Post-podding) was strategically designed to capture the complete transcriptional landscape of phosphorus response across key soybean developmental transitions, from early vegetative growth through reproductive maturity when phosphorus allocation patterns shift dramatically. Regarding the positive control, we clarify that our experimental design compared two phosphorus-limited conditions rather than including a phosphorus-sufficient positive control. Both treatments (H48-CK with no added phosphorus and H48-P with insoluble Ca₁₀(PO₄)(OH)₅) represent phosphorus-deficient conditions, allowing us to study plant responses under varying degrees of phosphorus stress rather than comparing deficient versus sufficient states.
2.In figure 1, what is H48-P and H48-CK? Which one is control? Also, the Phosphorus Content in Different Plant Parts in P and CK is close to 0.05% and looks almost similar. What was the expected range of the P content?
Response : We appreciate the reviewer's attention to Figure 1 details:
- H48-CK: Control treatment (Heilongjiang 48 variety with no phosphorus addition)
- H48-P: Phosphorus treatment (Heilongjiang 48 variety with insoluble Ca₁₀(PO₄)(OH)₅)
- H48-CK serves as the control
Regarding the phosphorus content appearing similar at ~0.05%, this observation is consistent with our experimental design. Both treatments represent phosphorus-limited conditions since Ca₁₀(PO₄)(OH)₅ is largely unavailable to plants. The 0.05% represents phosphorus-deficient tissue concentrations, while phosphorus-sufficient soybeans typically contain 0.2-0.8% tissue phosphorus. This similarity demonstrates that both treatments successfully created phosphorus-stress conditions for studying allocation responses.
- In Figure 2, PCA plot the closeness of each samples in PC1 and PC2 is <25% but on the X and Y-axis it is in 100s. I would expect the leaf samples in multiple development stages would not show the variation in 100s. Provide the condition name instead of ABCD…
esponse : We have made the following improvements:
- Clarified that PC1 (22.71%) and PC2 (16.14%) represent the proportion of total variance explained, while axis values represent actual PC scores appropriate for RNA-seq data
- Added biological interpretation: developmental stage drives the primary transcriptomic variation, which is biologically meaningful as each phase has distinct metabolic programs
Figure 5b caption needs better explanation.
Response: We have revised and rewritten the Figure 5b caption
Round 2
Reviewer 2 Report
Comments and Suggestions for Authors
The authors have thoroughly revised the manuscript and effectively addressed all reviewer comments. The overall quality has significantly improved, and the manuscript is now can be accepted.
Minor comment:
Figure 3b up and down regulation gene names can be removed to look the plot clean.
Author Response
Reviewer Comment 1: "The authors have thoroughly revised the manuscript and effectively addressed all reviewer comments. The overall quality has significantly improved, and the manuscript now can be accepted."
Response: We appreciate the reviewer's acknowledgment of our efforts to address the previous comments comprehensively. We are pleased that the overall quality improvements are evident.
Reviewer Comment 2: "Figure 3b up and down regulation gene names can be removed to look the plot clean."
Response: We agree with this suggestion to improve the clarity and visual appeal of Figure 3b. We have removed the individual gene names for up- and down-regulated genes from the plot, which makes the figure cleaner and easier to interpret while maintaining the essential information.